# Comparing pregnancy and pregnancy outcome rates between adolescents with and without pre-existing mental disorders

Nakyung Jeon[1,2]*, Yasser Albogami[3], Sun-Young Jung[4], Regina Bussing[5], Almut G. Winterstein[6]

1 Pusan National University College of Pharmacy, Busan, Republic of Korea, 2 Research Institute for Drug Development, Pusan National University, Busan, Republic of Korea, 3 Department of Clinical Pharmacy, College of Pharmacy, King Saudi University, Riyadh, Saudi Arabia, 4 College of Pharmacy and Department of Global Innovative Drugs, Chung-Ang University, Seoul, Republic of Korea, 5 Department of Psychiatry, University of Florida College of Medicine, Gainesville, FL, United States of America, 6 Department of Pharamceutical Outcomes and Policy, Department of Epidemiology, and Center for drug Evaluation and Safety, University of Florida, Gainesville, FL, United States of America

* nakyung.jeon@pusan.ac.kr

**Data Availability Statement:** Data for these analyses were made available to the authors through third-party license from Merative

## Abstract

### Background

There are limited population-based data on the role of mental disorders in adolescent pregnancy, despite the presence of mental disorders that may affect adolescents' desires and decisions to become pregnant.

### Objective

This study aimed to examine the relationship between specific types of mental disorders and pregnancy rates and outcome types among adolescents aged 13–19 years, using single-year age groups.

### Methods

We conducted a retrospective cohort study using data from the Merative™ MarketScan Research Databases. The study population consisted of females aged 13–19 years with continuous insurance enrollment for three consecutive calendar years between 2005 and 2015. Pregnancy incidence rates were calculated both overall and within the different categories of mental disorders. The presence of mental disorders, identified through diagnosis codes, was classified into 15 categories. Pregnancy and pregnancy outcome types were determined using diagnosis and procedure codes indicating the pregnancy status or outcome. To address potential over- or underestimations of mental disorder-specific pregnancy rates resulting from variations in age distribution across different mental disorder types, we applied age standardization using 2010 U.S. Census data. Finally, multivariable logistic regression models were used to examine the relationships between 15 specific types of mental disorders and pregnancy incidence rates, stratified by age.

MarketScan Research Database, a commercial data provider in the United States. As such, the authors cannot make these data publicly available due to data use agreement. Other researchers can access these data by purchasing a license through Merative MarketScan Research Database. Inclusion criteria specified in the Methods section would allow other researchers to identify the same cohort of patients we used for these analyses. Interested individuals may see "https://www.merative.com/real-world-evidence" for more information on accessing Merative MarketScan Research Database. The authors did not have any special access privileges other authors would not have.

**Funding:** This work is supported by Chonnam National University Hospital Biomedical Research Institute (BCRI202109-85), awarded to NJ. The funders had no role in study design, data collection and analysis, decision to publish, or preparation of the manuscript.

**Competing interests:** The authors have declared that no competing interests exist.

## Results

The age-standardized pregnancy rate among adolescents diagnosed with at least one mental disorder was 15.4 per 1,000 person-years, compared to 8.5 per 1,000 person-years among adolescents without a mental disorder diagnosis. Compared to pregnant adolescents without a mental disorder diagnosis, those with a mental disorder diagnosis had a slightly but significantly higher abortion rate (26.7% vs 23.8%, P-value < 0.001). Multivariable logistic regression models showed that *substance use-related disorders* had the highest odds ratios (ORs) for pregnancy incidence, ranging from 2.4 [95% confidence interval (CI): 2.1–2.7] to 4.5 [95% CI:2.1–9.5] across different age groups. Overall, *bipolar disorders* (OR range: 1.6 [95% CI:1.4–1.9]– 1.8 [95% CI: 1.7–2.0]), *depressive disorders* (OR range: 1.4 [95% CI: 1.3–1.5]– 2.7 [95% CI: 2.3–3.1]), *alcohol-related disorders* (OR range: 1.2 [95% CI: 1.1–1.4]– 14.5 [95% CI: 1.2–178.6]), and *attention-deficit/conduct/disruptive behavior disorders* (OR range: 1.1 [95% CI: 1.0–1.1]– 1.8 [95% CI: 1.1–3.0]) were also significantly associated with adolescent pregnancy, compared to adolescents without diagnosed mental disorders of the same age.

## Conclusion

This study emphasizes the elevated rates of pregnancy and pregnancy ending in abortion among adolescents diagnosed with mental disorders, and identifies the particular mental disorders associated with higher pregnancy rates.

## Introduction

Mental disorders are relatively common among adolescents, particularly females, and are a significant clinical and public health concern. In 2020, the prevalence of persistent depressive feelings among U.S. high school students was 42.0%, with a higher prevalence in females than in males (57% vs 29%) [1]. The percentage of female students who have? experienced persistent feelings of sadness or hopelessness has increased dramatically from 36% to 57% over the last decade, whereas there has only been an 8% increase in the rate among males during the same period [1]. Investigating the role of mental health problems in female-specific health outcomes is critical.

Adolescents with pre-existing mental health conditions may have inadequate knowledge, resources, and psychosocial skills to maintain healthy relationships and avoid unintended pregnancies [2]. Recent literature has demonstrated a significant relationship between mental health and sexual risky behaviors [3]. Ambivalent pregnancy desire, poor contraceptive behavior, lack of contraceptive knowledge, early ages of first intercourse, more sexual partners, and low self-efficacy have all been associated with several mental disorders [4–6]. Although limited, studies have demonstrated a lack of utilization of women's health services and medical care-seeking behaviors among mentally ill women [7–9]. In 2011, the rate of unintended pregnancies in the United States was 48%, with a higher rate (75%) among adolescents.

Understanding the epidemiology of pregnancy in adolescents with mental disorders is a critical step toward understanding the psychological and behavioral effects of mental disorders on pregnancy desires, disparities, and health outcomes [10, 11]. However, there is limited epidemiological data available to estimate the pregnancy rate among adolescents with a pre-

existing mental disorder. As an initial effort to improve women's health in vulnerable populations, this study aimed to identify patterns of adolescent pregnancy rates across different types of mental disorders among privately insured females aged 13–19 years old.

## Methods

### Data source

This study used healthcare claims data obtained from the Merative™ MarketScan Research Databases [12]. This database provides details on reimbursed health services, including medical encounters and drugs dispensed in outpatient pharmacies for patients in approximately 150 employer-sponsored insurance plans. Because the database includes claims from many private insurers and has very wide geographic coverage, it has been frequently used in analyses of healthcare utilization as a data source representing privately insured individuals [12–14]. A retrospective cohort of females aged 13–19 years between 2006 and 2014 was established. The inclusion criteria for the cohort was as follows: continuous enrollment for at least three years between 2005 and 2015, allowing for eligibility periods such as 2005–2007, 2006–2008 and 2013–2015.

### Pregnancy and pregnancy outcome type ascertainment

Pregnancy episodes were identified based on two or more inpatient or outpatient encounters, indicating pregnancy, antenatal care, or delivery with or without specified pregnancy outcomes. The identified pregnancies were categorized into pregnancy outcome types, namely abortion, ectopic pregnancy, stillbirth, and live birth, using the International Classification of Diseases, Ninth Revision, Clinical Modification (ICD-9-CM), the Current Procedural Terminology (CPT), and the Healthcare Common Procedure Coding System (HCPCS) codes assigned to inpatient or outpatient encounter (S1 Table). Pregnancies with an unspecified outcome were also considered, and were defined as pregnancies identified by diagnoses or procedure codes of prenatal visits without any diagnoses indicating the other four pregnancy outcome types. The conception date for each pregnancy episode was estimated based on previously validated algorithms, using five pregnancy outcome types [15–17]. Briefly, the algorithm assigned fixed gestational ages (GA) for each pregnancy endpoint: 273 days for live birth, 196 days for stillbirth, 70 days for spontaneous/induced abortions, and 56 days for ectopic pregnancy. For "pregnancy with unspecified outcome," a GA of 55 days was assigned based on the median time between the estimated last menstrual period (LMP) and the first pregnancy care claim for other episodes with a specified outcome [17]. Using the estimated conception date, pregnancy rates were calculated for all females aged 13–19 years in each year from 2006 to 2014. Of note, two consecutive years of data (2006–2007, 2007–2008, and 2014–2015) were assessed to determine pregnancies that began in the second year of the three years, using the aforementioned algorithm, which estimates the conception date of pregnancy episodes in claim-based healthcare data.

### Mental disorder ascertainment by mental disorder type

For each pregnancy, the year preceding the conception year was used to ascertain diagnoses of mental disorders. For this reason, three years of continuous eligibility were required. At least one inpatient or outpatient encounter with a mental disorder diagnosis as the primary diagnosis was required to identify a patient with a mental disorder. An overview of the study design and designation of the study population as having a pre-existing mental disorder and pregnancy episode (i.e., the outcome) is illustrated in Fig 1.

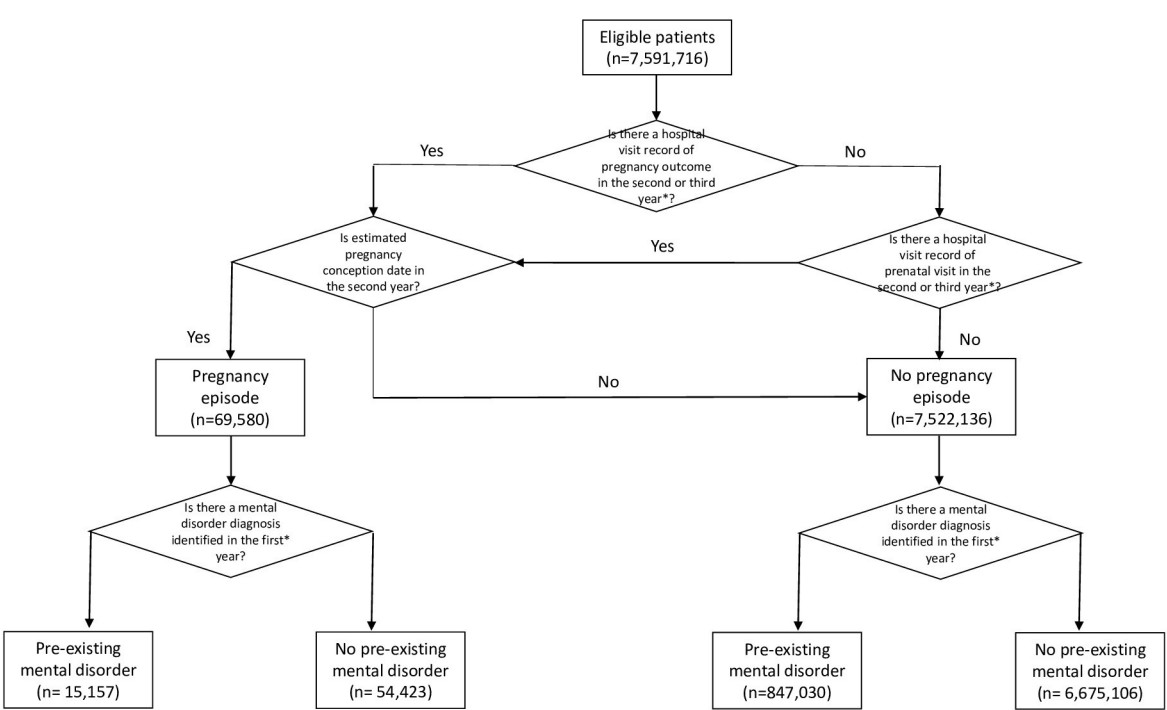

**Fig 1. Overview of study design.** The designation of the study population to having a pre-existing mental disorder and pregnancy episode to construct the binary variables (yes/no) is illustrated.

The Clinical Classification Software (CCS) provided by the Agency for Healthcare Research and Quality (CCS650–670) was modified to categorize each mental disorder-related diagnosis code into 15 disorder types: *Adjustment Disorders, Alcohol-Related Disorders, Anxiety Disorders, Attention-Deficit/Conduct/Disruptive Behavior Disorders, Bipolar Disorders, Delirium/ Dementia/Amnestic/Other Cognitive Disorders, Depressive Disorders, Developmental Disorders, Disorders Usually Diagnosed in Infancy, Childhood, or Adolescence Impulse Control Disorders, Miscellaneous Mental Disorders, Personality Disorders, Schizophrenia and Other Psychotic Disorders, Substance-Related Disorders, and Suicide Attempt and Intentional Self-inflicted Injury* [18]. The codes used in the modified CCS are available in S2 Table. Patients were assigned to multiple categories if they had two or more visits for the primary diagnoses of different types of mental disorder.

## Statistical analyses

Given the anticipated variations in mental disorder prevalence with age, it is important to address the potential underestimation or overestimation of pregnancy rates due to differences in age distribution among mental disorder groups. To mitigate the influence of age distribution on pregnancy rate estimation, age standardization was employed by aligning the estimated pregnancy rate in each mental disorder group with the 2010 U.S. Census age distribution [19]. To begin, the incidence rates of pregnancy were calculated for 15 different mental disorder groups, thereby determining the number of pregnancies within each mental disorder group. Next, the age distribution derived from the U.S. Census population was

applied to each mental disorder group to account for the diverse age distributions among the mental disorder groups and the U.S. Census population. Through this process, age-adjusted pregnancy rates were estimated for each mental disorder group, enabling a meaningful comparison of pregnancy rates across different mental disorder types while assuming a standardized age distribution across the mental disorder groups.

The distributions of the five pregnancy outcome types among pregnant adolescents were presented as counts and percentages with and without at least one mental disorder diagnosis. Differences in the rates of each pregnancy outcome type between adolescents with and without at least one mental disorder diagnosis were assessed using the Chi-square test. Finally, multivariable logistic regressions were used to identify the relationships between the types of mental disorders and the incidence of adolescent pregnancy stratified by age. In the logistic regression model, a binary coding approach was used to create separate dummy variables for each mental disorder category. This was used to assess the individual effects of each disorder while accounting for potential overlapping effects. However, it is important to note that the model did not explicitly consider the interactions or combined effects between multiple concurrent mental disorders. Instead, the focus was solely on examining the main effects of mental disorder variables on pregnancy events.

This study involved de-identified, aggregate data and was not subject to Institutional Review Board approval at the University of Florida. All methods were performed in accordance with the relevant guidelines and regulations of Merative™ MarketScan Research Databases [12].

Alpha was set to 0.05 as the significance level for all statistical analyses. SAS software version 9.4. was used for all analyses (SAS institute Inc., Cary, NC, USA).

## Results

The overall prevalence of mental disorders in female adolescents increased gradually from 9.7% in 2005 to 14.7% in 2013. *Depressive Disorders*, *Anxiety Disorders and Attention-deficit/ Conduct/Disruptive Behavior Disorders* were the three most prevalent mental disorders (Table 1). Approximately 95% of *Suicide Attempt and Intentional Self-inflicted Injury* diagnoses were concurrent with a diagnosis of *Depressive Disorder* (∼90%), *Anxiety Disorder* (∼62%), or *Attention-deficit/Conduct/Disruptive Behavior Disorder* (∼31%) in the same calendar year. Approximately 80% of *Personality Disorders*, *Schizophrenia and Other Psychotic Disorder* and over 60% of *Bipolar*, *Impulse Control* or *Substance-related Disorders* were concurrent with at least one of the three most prevalent disorders.

As shown in Table 2, approximately nine out of a thousand female adolescents experienced pregnancy in a given year. The age-standardized pregnancy rates were 15.4 (95% CI: 15.1–15.6) and 8.5 per 1,000 person-years (95% CI: 8.4–8.6) for adolescents with and without mental disorders, respectively. Live births accounted for more than half of the pregnancies (68.1%), followed by abortions (24.4%) and unspecified pregnancy outcomes (6.6%). Ectopic pregnancies and stillbirths contributed to 1% of pregnancy incidence. Compared to pregnant adolescents without mental disorders, those with mental disorders had a slightly but significantly higher abortion rate (26.7% vs 23.8%, P-value < 0.001) and a lower live birth rate (65.6% vs 68.8%, P-value < 0.001), as shown in Table 2.

Overall, age-standardized pregnancy rates in female adolescents with at least one mental disorder diagnosis were relatively high regardless of mental disorder type, with the exception of *Developmental Disorders* (7.4 events/1,000person-year, 95% CI: 6.1–8.7) and *Disorders Usually Diagnosed in Infancy, Childhood, or Adolescence (DICA)*, such as autism spectrum disorders or tic disorders (6.5 events/1,000person-years, 95% CI: 5.4–7.6). Female adolescents with

**Table 1. Crude and age–standardized pregnancy rates per 1,000 person-year of females aged 13–19 years, overall and according to mental disorder type.**

| | Numerator | Denominator | Crude rate | Age-standardized rate | 95% Confidence intervals |
|---|---|---|---|---|---|
| Overall | 69,580 | 7,591,716 (100%) | 9.2 | 9.3 | 9.2–9.4 |
| No mental disorder | 54,423 | 6,729,529 (88.6%) | 8.1 | 8.5 | 8.4–8.6 |
| With any mental disorder | 15,157 | 862,187 (11.4%) | 17.6 | 15.4 | 15.1–15.6 |
| *Substance-Related Disorders* | 1,727 | 28,429 (0.4%) | 60.7 | 42.2 | 39.9–44.5 |
| *Alcohol-Related Disorders* | 913 | 19,054 (0.3%) | 47.9 | 34.5 | 30.5–38.6 |
| *Suicide Attempt and Intentional Self-inflicted Injury* | 755 | 19,103 (0.3%) | 39.5 | 32.5 | 30.0–34.9 |
| *Bipolar Disorders* | 2,622 | 78,708 (1.0%) | 33.3 | 26.5 | 25.5–27.6 |
| *Personality Disorders* | 216 | 6,795 (0.1%) | 31.8 | 24.3 | 20.9–27.6 |
| *Schizophrenia and Other Psychotic Disorders* | 350 | 12,236 (0.2%) | 28.6 | 23.0 | 20.6–25.5 |
| *Depressive Disorders* | 6434 | 259,321 (3.4%) | 24.8 | 18.8 | 18.3–19.3 |
| *Impulse Control Disorders* | 145 | 7,620 (0.1%) | 19.0 | 18.1 | 15.1–21.1 |
| *Anxiety Disorders* | 4313 | 239,985 (3.2%) | 18.0 | 14.7 | 14.2–15.1 |
| *Miscellaneous Mental Disorders* | 1072 | 61,485 (0.8%) | 17.4 | 13.8 | 12.9–14.6 |
| *Adjustment Disorders* | 3102 | 200,626 (2.6%) | 15.5 | 14.2 | 13.7–14.7 |
| *Attention-Deficit/Conduct/Disruptive Behavior Disorders* | 3667 | 265,927 (3.5%) | 13.8 | 14.4 | 13.9–14.9 |
| *Delirium/Dementia/Amnestic/Other Cognitive Disorders* | 198 | 15,251 (0.2%) | 13.0 | 10.8 | 9.2–12.3 |
| *Developmental Disorders* | 127 | 20,790 (0.3%) | 6.1 | 7.4 | 6.1–8.7 |
| *Disorders Usually Diagnosed in Infancy, Childhood, or Adolescence* | 144 | 27,656 (0.4%) | 5.2 | 6.5 | 5.4–7.6 |

*Substance-related Disorders, Alcohol-related Disorders, Suicide Attempt and Intentional Self-inflicted Injury, Personality Disorders, and Bipolar Disorders* had the highest age-standardized pregnancy rates ranging from 24.3 (95% CI: 20.9–25.5) to 42.2 (95% CI: 39.9–44.5) per 1,000 person-years (Table 1).

Pregnancy rates increased with age from 0.2 to 25.0 per 1000 person-years. Fig 2 displays forest plots illustrating the Odds Ratios (OR) with 95% CIs for adolescent pregnancies in cohorts stratified by patient age from 14 to 19 years. Multivariable analyses in age-stratified cohorts indicated that patients with certain mental disorders had a significantly higher adolescent pregnancy incidence regardless of age, including *Substance-related Disorders, Bipolar Disorders and Depressive Disorders*.

Overall, *Substance-related Disorders* (OR: 2.44 [95% CI: 2.33–3.02]), *Bipolar Disorders* (OR: 1.77 [95% CI: 1.71–1.85]), *Depressive Disorders* (OR: 1.57 [95% CI: 1.53–1.61]), *Alcohol-related Disorders* (OR: 1.42 [95% CI: 1.34–1.53]), *Attention-deficit/Conduct/Disruptive Behavior Disorders* (OR: 1.23 [95% CI: 1.19–1.27]), *Adjustment Disorders* (OR: 1.21 [95% CI: 1.17–1.25]), *Suicide Attempt and intentional Self-inflicted Injury* (OR: 1.17 [95% CI: 1.09–1.26]), *Miscellaneous*

**Table 2. Distribution of outcome types according to the presence of pre-existing mental disorder in females aged 13–19 years.**

| | The Number of Pregnancy | | | |
|---|---|---|---|---|
| Pregnancy outcome types | Without any Mental Disorder Diagnosis | With at least one mental disorder diagnosis | P-value | All |
| Live birth | 37,432 (68.78%) | 9,939 (65.57%) | < 0.001 | 47,371 (68.08%) |
| Abortion | 12,935 (23.77%) | 4,047 (26.70%) | < 0.001 | 16,982 (24.41%) |
| Unspecified pregnancy outcomes | 3,530 (6.49%) | 1,046 (6.90%) | 0.07 | 4,576 (6.58%) |
| Ectopic pregnancy | 252 (0.46%) | 58 (0.38%) | 0.18 | 310 (0.45%) |
| Stillbirth | 274 (0.50%) | 67 (0.44%) | 0.34 | 341 (0.49%) |
| Total | 54,423 (100%) | 15,157 (100%) | | 69,580 (100%) |

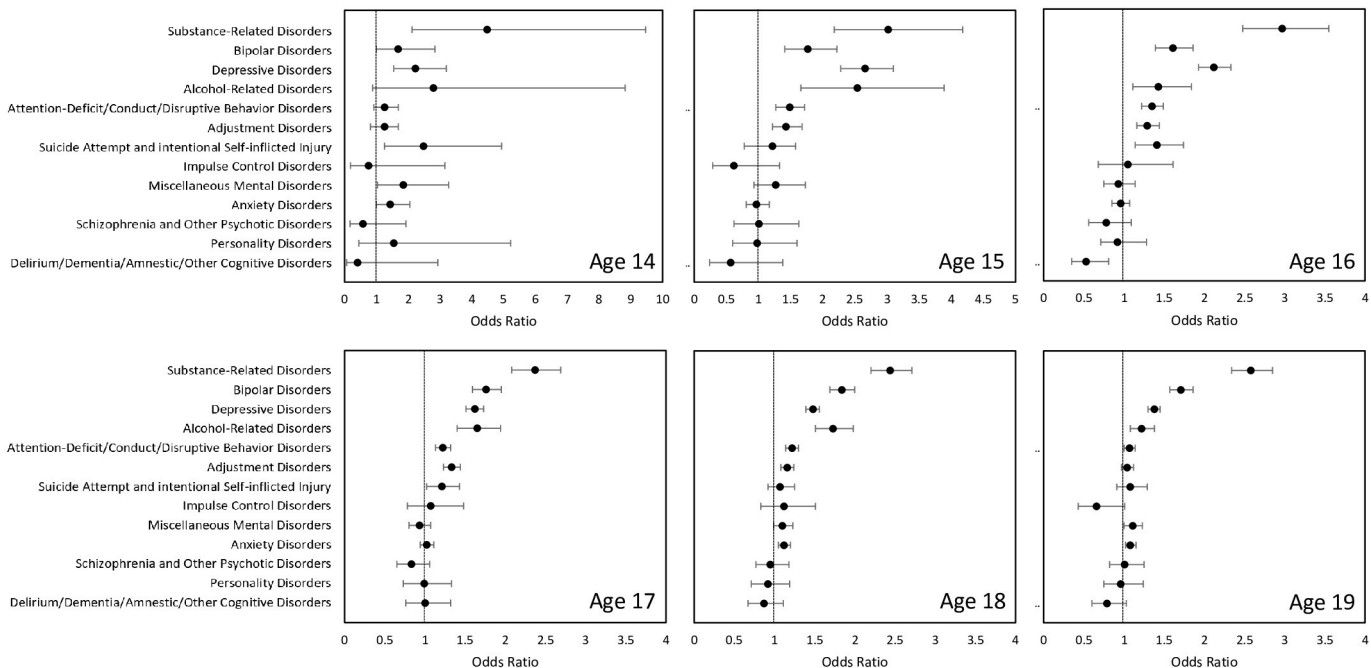

\* Statistically significant with alpha = 0.05
\*\* Developmental Disorders/Disorders Usually Diagnosed in Infancy, Childhood, or Adolescence were not included in the age-stratified analysis due to small sample size

**Fig 2. Age-stratified odds ratios of becoming pregnancy among adolescents with a mental disorder diagnosis.** The presence of mental disorder diagnosis and its type was evaluated in the first year of the three-year continuous eligibility period. The odds ratios were calculated in a multivariable logistic regression for a certain age group. In each logistic regression model, patients with one of the 15 types of mental disorders were evaluated together for their likelihood of having a pregnancy episode, and patients with no mental disorder were used as a reference group.

*Mental Disorders* (OR: 1.07 [95% CI:1.01–1.13]), *Anxiety Disorders* (OR: 1.05 [95% CI: 1.02–1.09]) were associated with higher adolescent pregnancy incidence (S1 Fig). However, patients with *Disorders Usually Diagnosed in Infancy*, *Childhood, or Adolescence* (OR: 0.52 [95% CI: 0.46–0.60]) or *Developmental Disorders* (OR: 0.73 [95% CI: 0.63–0.83]) exhibited lower pregnancy rates when compared to adolescents without any diagnosed mental disorders (S1 Fig).

## Discussion

This was the first and largest retrospective cohort study to estimate pregnancy rates for different pre-existing mental disorders among adolescents. Our analysis of this representative sample of private insurance enrollees demonstrated that adolescents with pre-existing mental disorders had disproportionately higher pregnancy rates than their peers without such disorders.

Adolescents with mental disorders characterized by risk-taking behaviors including *Substance-related Disorders*, A*lcohol-related Disorders*, *Bipolar Disorder*, and *Attention-deficit/ Conduct/Disruptive Behavior Disorders* had relatively high pregnancy rates. The results suggest that a large number of pregnancies in adolescents may be influenced by the effects of underlying psychiatric conditions on mood, behavior, and decision-making or treatment.

While the reasons for high adolescent pregnancy rates in individuals with mental disorders were not examined in this study, our study confirmed that mental disorders and pregnancy frequently coexist in adolescents and the pregnancy rates were higher among adolescents with certain mental disorder types.

The coexistence of mental disorders and pregnancy among adolescents underscores the need for public health interventions that extend beyond the healthcare system. These include promoting mental health awareness, improving access to mental health services, enhancing sexual and reproductive health education, and providing comprehensive and effective support for pregnant adolescents with mental disorders.

Pre-existing mental disorders may continue or worsen during pregnancy period. The management of mental disorders during and after pregnancy is challenging and requires specialized healthcare and social support. Some adolescents diagnosed with mental disorders may already be taking psychotropic medications. Hence, the introduction of pregnancy adds complexity to their healthcare needs because of the need to balance between the effectiveness and safety of the psychotropic medications. In addition, some mental disorders or their pharmacotherapy can affect the hypothalamic–pituitary–gonadal axis, leading to anovulatory cycles and menstrual disturbances [20–23]. These conditions may also interact with hormonal contraceptives, reducing effectiveness of contraceptive methods or the treatment.

Some mental disorders are also associated with poor medication adherence. Children born to mothers with untreated or poorly managed mental disorders may be at higher risk of developmental and behavioral problems. Mental disorders may also lead to poor prenatal care attendance and poor maternal health. Our study indicated that pregnant adolescents diagnosed with at least one mental disorder diagnosis had a higher abortion rate than those without a mental disorder diagnosis, a finding that partially supports the possibility of an increased likelihood of pregnancy complications and negative outcomes.

Our results emphasize the importance of public health interventions that can reduce unintended pregnancies in the context of mental disorder management. This is particularly relevant for adolescents who may face unique challenges due to their young age, severity of mental health, or pharmacotherapy, which can potentially harm the fetus. Although validated programs and evidence to support interventions for pre-conception health exist, they are directed toward encouraging safer-sex practices in the general adolescent population, and the benefits of these interventions for adolescents with mental disorders await further evaluation [24, 25].

One of the main findings of our study was that adolescents diagnosed with substance use disorder had a pregnancy rate more than double than that of adolescents without any mental disorder diagnosis. In response to the opioid crisis in the United Sates, the understanding of the epidemiology and evidence-based treatment of opioid use disorder has evolved over the past several decades. However, research or programs designed to improve outcomes in opioid use disorders are almost entirely absent for adolescents [26]. Given the unique importance for women of childbearing age to avoid substance use and for women with substance use disorders to prevent pregnancy, screening (i.e., drug test and pregnancy test) or the provision of relevant resources should be accessible to women to reduce the risk of substance use during pregnancy.

Interestingly, we found *Depressive Disorders* played a larger role in adolescent pregnancy rate at younger age. The ORs for adolescent pregnancies at ages 14–16 years were significantly higher than those at ages 18–19. While there are limited data specifically documenting the demographics of pregnant women with pre-pregnancy depression, related findings regarding higher rates of sexual initiation at a younger age, misuse of contraception, unintended pregnancy, and induced abortion among women with depression may support our findings [10, 27, 28].

In contrast, *Disorders Usually Diagnosed in Infancy*, *Childhood*, *or Adolescence* and *Developmental Disorders* such as intellectual disability (ID) exhibited a decreased adolescent pregnancy rate in our analytic cohort. To our knowledge, no previous study has directly compared the incidence of pregnancy between adolescents with and without ID. Further evaluation of pregnancy intentions in adolescents with ID will be particularly meaningful given the known vulnerability of children with ID in becoming victims of sexual abuse [29].

This study had several limitations that should be acknowledged when interpreting our findings. First, this study could not differentiate between unintended and intended pregnancies among those identified pregnancies in adolescents. Over 70%– 80% of adolescent pregnancies are commonly recognized as unintended, and the presence of mental disorders may potentially affect the intention to become pregnant [30, 31]. Among the individuals who do not intend to become pregnant, having certain mental disorders may increase or decrease the likelihood of conception, which may be affected by impulsivity, risky behavior, or difficulties in decision making and in considering the consequences of their actions [32] (S2 Fig). Future studies evaluating pregnancy intention as a mediator on the association between a mental disorder and adolescent pregnancy are warranted.

Second, despite our efforts to identify all conceptions and their timing regardless of pregnancy outcome type, we may have underestimated pregnancy rates due to failure to capture miscarriages and non-reimbursed abortion, and misclassification of gestational age. However, when interpreting the results, the focus should be on the pattern of pregnancy rates due to mental disorders, as the extent of misclassification would not be expected to differ systematically across diagnoses. Third, the data comprised patients with three continuous years of private health insurance, implying that all included study populations had employed parents. Therefore, the study results and interpretations cannot be generalized to uninsured or government insured populations. Fourth, we included age as the only covariate in the multivariable logistic regression analysis after adjusting for the relationship between the type of mental disorder and adolescent pregnancy. However, differences in race, income, or geographic area among mental disorder types may exist, which can affect the estimation of adolescent pregnancy. Future studies that provide basis for adjusting for differences in age-specific adolescent pregnancy rates across groups defined by sex, race, ethnicity, geography, and other sociodemographic categories are warranted.

Lastly, while we do not discount a more specific designation of mental disorder diagnoses requiring more stringent criteria (e.g., at least two outpatient diagnoses of the same mental disorder), we emphasize the distinction between girls with some level of mental disorder and those who were never diagnosed with any mental disorder during our look-back period.

## Conclusion

With significantly higher adolescent pregnancy rates than their counterparts without diagnosed mental disorders, adolescents with certain mental disorders should be prioritized for evidence-based pre-conception health care.

## Supporting information

**S1 Table. Pregnancy-related ICD-9-CM codes.** The list of ICD-9-CM codes to indicate prenatal visits, pregnancy outcomes, or delivery.
(DOCX)

**S2 Table. The fifteen mental disorder categories.** The list of ICD-9-CM codes categorized by 15 mental disorders based on the Clinical Classifications Software (CCS) developed by the Agency for Healthcare Research and Quality (AHRQ).
(DOCX)

**S1 Fig. Age-adjusted odds ratios of becoming pregnant among adolescents with a mental disorder diagnosis.**
(TIF)

**S2 Fig. A directed acyclic graph illustrating the role of pregnancy intention as a mediator on the association between mental disorder and adolescent pregnancy.**
(TIF)

## Author Contributions

**Conceptualization:** Nakyung Jeon, Regina Bussing, Almut G. Winterstein.

**Formal analysis:** Nakyung Jeon.

**Methodology:** Nakyung Jeon, Yasser Albogami, Sun-Young Jung, Regina Bussing, Almut G. Winterstein.

**Resources:** Almut G. Winterstein.

**Supervision:** Nakyung Jeon, Almut G. Winterstein.

**Writing – original draft:** Nakyung Jeon.

**Writing – review & editing:** Nakyung Jeon, Yasser Albogami, Sun-Young Jung, Regina Bussing, Almut G. Winterstein.

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
