## [Decision Letter · Decision Letter 0]

25 Apr 2023

PONE-D-23-05808Comparing pregnancy and pregnancy outcome rates between adolescents with and without a pre-existing mental disorderPLOS ONE

Dear Dr. Jeon,

Thank you for submitting your manuscript to PLOS ONE. After careful consideration, we feel that it has merit but does not fully meet PLOS ONE’s publication criteria as it currently stands. Therefore, we invite you to submit a revised version of the manuscript that addresses the points raised during the review process.

We look forward to receiving your revised manuscript.

Kind regards,

Giuseppe Marano

Academic Editor

PLOS ONE

Reviewers' comments:

Reviewer's Responses to Questions

**Comments to the Author**

1. Is the manuscript technically sound, and do the data support the conclusions?

Reviewer #1: Partly

Reviewer #2: Yes

2. Has the statistical analysis been performed appropriately and rigorously? 

Reviewer #1: I Don't Know

Reviewer #2: I Don't Know

3. Have the authors made all data underlying the findings in their manuscript fully available?

Reviewer #1: Yes

Reviewer #2: No

4. Is the manuscript presented in an intelligible fashion and written in standard English?

Reviewer #1: No

Reviewer #2: Yes

5. Review Comments to the Author

Reviewer #1: Study Summary: Using a retrospective cohort study design, with data was derived from the 2005-2015 IBM Marketscan Commercial Claims Research Database, the authors of this manuscript sought to examine the association between a mental health diagnosis and rates of pregnancy among adolescents aged 13-19. Investigators performed age-standardized logistic regression models and determined that adolescents with a mental health disorder had nearly a two times increased rate of pregnancy compared to those of a similar age group without a mental health disorder. Those diagnosed with substance-use related disorders showed the most striking differences. Pregnancy outcome types were also observed among the two groups with results indicating that those with a mental health disorder have a greater rate of abortions than those without. Implications are geared towards interventions that reduce unplanned pregnancy. The scope of this work is highly beneficial for amplifying the role of preconception mental health research and clinical practice. The authors present a interesting analysis on a very important topic. Additional strengths include a strong data source, with results adding to a much-needed growing body of research. However, several areas of clarification require further description. Given the comments below regarding the statistical approaches used, I would also recommend that a statistician review this manuscript. Overall, the study findings are a bit hard to interpret because some of the data and methods have not been provided or explained. Additionally, there are grammatical errors and stigmatizing language that need attention. Please find detailed feedback below:

Abstract:

1. Major: The authors state they used a case cohort study design, however there is no description of how cases were identified. A case cohort is used when multiple case groups are defined from a cohort study, compared to a control series from the same cohort. These are often useful when the risk ratio is of greater interest than incidence rate ratio or when multiple outcomes are of interest. I think this study can simply be stated as a retrospective cohort study.

2. Minor: Please revise for grammatical errors.

3. Minor: Please note at what confidence level the ranges in the results were derived from.

Introduction:

1. Minor: Please revise for grammatical errors.

2. Major: Avoid the term mental health ‘issues’ as this can be stigmatizing.

3. Major: Please incorporate additional citations in statements derived from evidence-based sources

4. Major: The introduction is a bit confusing, and several points can either be elaborated upon or stated more clearly. As written. It is unclear if the exposure is mental health status or a mental health condition/diagnosis.

5. Major: The investigators state that this study was developed as a means to prevent unplanned pregnancy. However, unplanned pregnancy was not discussed in this population at any other time in the introduction. This should be removed, or the introduction should be framed around the prevalence of unplanned pregnancy among persons diagnosed with a mental health disorder.

Methods:

1. Minor: Please provide a brief description of the validated algorithms used to create the five pregnancy outcome types.

2. Minor: Please add sub-headers throughout this section to improve readability. It was a bit challenging to follow what the measure were, from the data source and the analytic section (ex: add a ‘Statistical Analyses” right before the discussion of how distributions were assessed).

3. Major: Please discuss how comorbidities were addressed in the selection of mental health disorder type. Many of these conditions co-occur with one another and the authors allude to this point later on in the manuscript.

4. Major: In your statistical analysis you describe use of stratification – were possible interactions assessed in these?

5. Minor: The procedures used for standardization can be more clearly described.

6. Major: I do not see other covariates listed (outside of the age-based standardization). This is very concerning as no other descriptions of this population are provided. Furthermore, there are no confounders listed and there are several factors that are related with both exposure and outcome and would need to be controlled along the casual pathway.

Results:

1. Major: Terminology about the study population can be strengthened. Use of the term, ‘girls’ vs ‘females’ vs ‘adolescents’ vs ‘women’ makes it hard to follow. Please pick one and use it consistently throughout. ‘Adolescence’ may be the most appropriate and gender-neutral descriptor. Alternatively, you can make a note in the introduction describing that you will use the terms interchangeably.

2. Minor: Please define a tic disorder?

3. Minor: Please include confidence intervals when presenting your data.

4. Minor: Which exposure group does the 7.4 per 1000 and 6.5 per 1,000 finding pertain to? (Page 5)

5. Major: Please save all language that is geared towards interpreting the results for the discussion section (ex: ‘Consistent with previous studies’, ‘as expected’).

6. Major: Comorbidities are presented in the analyses and therefore should be discussed in the methods.

7. Minor: A hypothesis was not presented originally in the introduction and therefore ‘as expected’ is not appropriate in this section nor throughout the manuscript.

8. Major: The organization of study findings can be revised dramatically to increase readability. Please present prevalence’s and then measures of effect. As written, the outcome is unclear.

9. Minor: Physician-diagnosed mental disorders is used for the first time here. Either make consistent throughout or remove from this section.

10. Table 2: The exposure group of ‘adolescents with at least one mental disorder diagnosis’ is clear. This is the language that should be used throughout.

11. Table 2: Please round to 2 decimal places.

Discussion:

1. Major: It is unclear what the intent of paragraph 2 is, and I am not certain that the implications described here are justified based on the findings.

2. Minor: Some of the articles chosen to compare study findings may need revision. For this manuscript, I do not think treatment seeking women with opioid use disorder is a good group to use to present results in light of other work.

3. Major: This section should be revised to avoid stigmatizing language and to more aligned with harm reduction approaches. Intervention approaches should also be mindful of this unique age group. Particularly on page 3 use of the term ‘healthy peers’ is very jarring and should not be used at all.

4. Minor: What does DICA stand for?

5. Major: Again, I think before the discussion can be framed around unplanned pregnancy the introduction needs to be clear about this as the goal of this investigator. Also, while pregnancy intent is not something that could possibly be obtained from the data source, it may be helpful to create a directed acyclic graph to make sense of some of these relationships. If anything, it may be that unplanned pregnancy is a mediator.

6. Minor: STI’s do not contribute to unplanned pregnancy (page 4).

7. Minor: Please be consistent with what is the outcome vs outcome type throughout.

Reviewer #2: Little is known about the association between mental health and adolescent pregnancy. In this manuscript, the authors report on the findings of a retrospective case-cohort study of females at age 13-19. This study, therefore, makes a valuable contribution to the literature. Below I give some suggestions to improve the paper.

• It is unclear how the retrospective cohort of females was established (first sentence under methods). Perhaps it would be better to describe the source of data, and then follow this up with the description of the retrospective cohort. In describing the cohort, the text in the parentheses (i.e., eligibility period…..) in the first sentence can be deleted.

• It is unclear if the database includes information on sociodemographic characteristics of privately insured people (e.g., race, income, etc.). Including these in the model would be useful in assessing whether these characteristics are also associated with the outcomes. These data would also be useful in understanding how the study sample may differ from the wider population. This could also be highlighted as a limitation if the data are not available.

• It would have been useful to show whether the likelihood of pregnancy was different for those with comorbid mental health disorders compared to those with only one mental disorder diagnosis. This could have implications for the targeting of prevention interventions.

Minor edits

• The second paragraph in the discussion does not seem to add much and can be deleted.

• The manuscript needs a careful read through for grammar and spelling errors. Below I highlight just a few from the first two paragraphs

- Para 1, First sentence: Mental health disorders in “adolescence”

- Para 1, Second sentence: “The 2020 national prevalence…. in high schools in the United States was 42%”

- Para 1, Third sentence: “hopelessness has increased dramatically”

- Para 2, First sentence: “adolescent mental”

- Para 2, Second sentence: “An existing mental health problem/disorder”

6. PLOS authors have the option to publish the peer review history of their article (what does this mean?). If published, this will include your full peer review and any attached files.

Reviewer #1: **Yes: **Brian W Jack

Reviewer #2: No

---

## [Author Response · Author response to Decision Letter 0]

17 Aug 2023

Reviewer #1: Study Summary: Using a retrospective cohort study design, with data was derived from the 2005-2015 IBM Marketscan Commercial Claims Research Database, the authors of this manuscript sought to examine the association between a mental health diagnosis and rates of pregnancy among adolescents aged 13-19. Investigators performed age-standardized logistic regression models and determined that adolescents with a mental health disorder had nearly a two times increased rate of pregnancy compared to those of a similar age group without a mental health disorder. Those diagnosed with substance-use related disorders showed the most striking differences. Pregnancy outcome types were also observed among the two groups with results indicating that those with a mental health disorder have a greater rate of abortions than those without. Implications are geared towards interventions that reduce unplanned pregnancy. The scope of this work is highly beneficial for amplifying the role of preconception mental health research and clinical practice. The authors present a interesting analysis on a very important topic. Additional strengths include a strong data source, with results adding to a much-needed growing body of research. However, several areas of clarification require further description. Given the comments below regarding the statistical approaches used, I would also recommend that a statistician review this manuscript. Overall, the study findings are a bit hard to interpret because some of the data and methods have not been provided or explained. Additionally, there are grammatical errors and stigmatizing language that need attention. Please find detailed feedback below:

Thank you for taking the time to review our manuscript. We appreciate your feedback and are grateful for your recognition of the study's strengths, including the robust data source and the implications for pre-conception mental health research and clinical practice. In response to your comments, we have carefully revised the manuscript to provide additional clarification of our study methods and data presentation to ensure that the findings are communicated clearly. We also agree with your suggestion to involve a statistician in reviewing the manuscript to verify the appropriateness of our statistical approach. Regarding the grammatical errors and stigmatizing language, we have made necessary revisions to enhance the manuscript's clarity and accuracy. Thank you again for your constructive comments, which have helped us improve the manuscript's quality.

Abstract:

1. Major: The authors state they used a case cohort study design, however there is no description of how cases were identified. A case cohort is used when multiple case groups are defined from a cohort study, compared to a control series from the same cohort. These are often useful when the risk ratio is of greater interest than incidence rate ratio or when multiple outcomes are of interest. I think this study can simply be stated as a retrospective cohort study.

We acknowledge the discrepancy between our stated study design as a case-cohort study and the absence of a clear description of how the cases within the sub-cohort were identified. We agree with your suggestion that this study may be more accurately described as a retrospective cohort study. As such, we changed the study design statement to “retrospective cohort study”. (Line 8)

2. Minor: Please revise for grammatical errors.

We have carefully revised the abstract to correct all grammar errors.

3. Minor: Please note at what confidence level the ranges in the results were derived from.

We used a 95% confidence level to calculate the confidence intervals in our results and have indicated this information in the abstract. (Lines 27 – 32)

Introduction:

1. Minor: Please revise for grammatical errors.

We have carefully revised the introduction to correct all grammar errors.

2. Major: Avoid the term mental health ‘issues’ as this can be stigmatizing.

We replaced “mental health issues” with “mental disorders”, “pre-existing mental health conditions”, mental health problems”, and “mentally ill” as appropriate.

3. Major: Please incorporate additional citations in statements derived from evidence-based sources

We have added the following five citations as appropriate in the introduction.

2. Olmsted AE, Markham CM, Shegog R, Ugueto AM, Johnson EL, Peskin MF, Emery ST, Baker KA, Newlin EW. Feasibility and Acceptability of Technology-supported Sexual Health Education Among Adolescents Receiving Inpatient Psychiatric Care. J Child Fam Stud. 2022;31(7):2050-2064. 

3. Ramrakha S, Caspi A, Dickson N, Moffitt TE, Paul C. Psychiatric disorders and risky sexual behaviour in young adulthood: cross sectional study in birth cohort. BMJ. 2000;29;321(7256):263-6. 

7. Berry MS, Sweeney MM, Dolan SB, Johnson PS, Pennybaker SJ, Rosch KS, Johnson MW. Attention-Deficit/Hyperactivity Disorder Symptoms Are Associated with Greater Delay Discounting of Condom-Protected Sex and Money. Arch Sex Behav. 2021;50(1):191-204. 

8. Verlenden JV, Bertolli J, Warner L. Contraceptive Practices and Reproductive Health Considerations for Adolescent and Adult Women with Intellectual and Developmental Disabilities: A Review of the Literature. Sex Disabil. 2019;37(4):541-557. 

9. Goyal S, Monsour M, Ko JY, Curtis KM, Whiteman MK, Coy KC, Cox S, Romero L. Contraception claims by medication for opioid use disorder prescription status among insured women with opioid use disorder, United States, 2018. Contraception. 2023;117:67-72.

4. Major: The introduction is a bit confusing, and several points can either be elaborated upon or stated more clearly. As written. It is unclear if the exposure is mental health status or a mental health condition/diagnosis.

We have revised the introduction entirely to provide more clarity on the exposure (=a mental disorder)

5. Major: The investigators state that this study was developed as a means to prevent unplanned pregnancy. However, unplanned pregnancy was not discussed in this population at any other time in the introduction. This should be removed, or the introduction should be framed around the prevalence of unplanned pregnancy among persons diagnosed with a mental health disorder.

We have revised the introduction as follows to frame the introduction around the unintended pregnancy among individuals with a mental disorder. (Lines 55-66)

“In 2011, the rate of unintended pregnancy in the United States was 48%, with a higher rate (75%) in adolescents. At present, studies on unintended pregnancy among adolescents with mental disorders are sparse. Alternatively, a meta-analysis of 11 studies reported that the rate of unintended pregnancy among mentally ill women (regardless of age) concluded that women diagnosed with mood, anxiety, psychotic, substance use, conduct or eating disorders have a 34% higher risk of unintended pregnancy than women without such a diagnosis. A Canadian study that examined the prevalence and characteristics of adolescent women who intended to become pregnant found that approximately 27% of the pregnancies were intended. Furthermore, these women were less likely to have experienced violence within the last two years or had used alcohol prior to their pregnancy. However, adolescent women who had intended to become pregnant were more likely to have used drugs before pregnancy.”

Methods:

1. Minor: Please provide a brief description of the validated algorithms used to create the five pregnancy outcome types.

The following statement was added to the methods. (Lines 98 – 103) 

“Briefly, the algorithm assigned fixed gestational ages (GA) for each pregnancy endpoint: 273 days for live birth, 196 days for stillbirth, 70 days for spontaneous/induced abortions, and 56 days for ectopic pregnancy. For "pregnancy with unspecified outcome," a GA of 55 days was assigned based on the median time between the estimated last menstrual period (LMP) and the first pregnancy care claim for other episodes with a specified outcome.”

2. Minor: Please add sub-headers throughout this section to improve readability. It was a bit challenging to follow what the measure were, from the data source and the analytic section (ex: add a ‘Statistical Analyses” right before the discussion of how distributions were assessed).

We have added the following subheadings to the method section: Data source, Pregnancy and pregnancy outcome ascertainment, Mental disorder ascertainment by mental disorder type, and Statistical Analyses

3. Major: Please discuss how comorbidities were addressed in the selection of mental health disorder type. Many of these conditions co-occur with one another and the authors allude to this point later on in the manuscript.

In our logistic regression model, we did not specifically account for patients with multiple concurrent mental disorders. Rather, we treated each mental disorder as a separate independent variable. Therefore, the interpretation of the model for a patient with multiple concurrent mental disorders would involve examining the individual effects of each disorder independently, without explicitly considering potential interactions or combined effects. To clarify, we have added the following paragraph to the manuscript. (Lines 152 – 158)

“In the logistic regression model, a binary coding approach was used to create separate dummy variables for each mental disorder category. This was used to assess the individual effects of each disorder while accounting for potential overlapping effects. However, it is important to note that the model did not explicitly consider the interactions or combined effects between multiple concurrent mental disorders. Instead, the focus was solely on examining the main effects of the mental disorder variables on pregnancy events.”

4. Major: In your statistical analysis you describe use of stratification – were possible interactions assessed in these?

Our analysis did not involve a formal assessment of the interaction between age and mental disorder type on the occurrence of pregnancy. Instead, we repeated the aforementioned logistic regression analyses within individual cohorts stratified by age to investigate how mental disorder type might exert different impacts on pregnancy occurrence across various age-groups. By utilizing this approach, we aimed to explore the potential variations in the association between mental disorders and pregnancy events within distinct age strata. 

5. Minor: The procedures used for standardization can be more clearly described.

We have provided an explanation of the age-standardization procedure as follows (Lines 134 – 146);

“Given the anticipated variations in mental disorder prevalence with age, it is crucial to address the potential underestimation or overestimation of pregnancy rates due to differences in age distributions among mental disorder groups. To mitigate the influence of age distribution on pregnancy rate estimation, age-standardization was employed by aligning the estimated pregnancy rate in each mental disorder group with the 2010 U.S. Census age-distribution. To begin, the incidence rates of pregnancy were calculated for 15 different mental disorders, thereby determining the number of pregnancies within each mental disorder group. Next, the age distribution derived from the U.S. Census population was applied to each mental disorder group to account for the diverse age distributions among the mental disorder groups and the U.S. Census population. Through this process, the age-adjusted pregnancy rates were estimated for each mental disorder group, enabling a meaningful comparison of pregnancy rates across different mental disorder types while assuming a standardized age distribution across the mental disorder groups.”

6. Major: I do not see other covariates listed (outside of the age-based standardization). This is very concerning as no other descriptions of this population are provided. Furthermore, there are no confounders listed and there are several factors that are related with both exposure and outcome and would need to be controlled along the casual pathway.

The objective of this study did not involve examining the causal pathway between mental disorder type and pregnancy incidence. Thus, the application or utilization of epidemiological methods for potential confounding or selection bias was not necessary, as they are not relevant to the research focus.

However, we do acknowledge that it is a study limitation that we only included “age” in our analysis as opposed to other sociodemographic information available in the data source. We added the following paragraph to address the limitation in the discussion section. (Lines 305 – 311)

“…, we included age as the only covariate in the multivariable logistic regression after adjusting for the relationship between the type of mental disorder and adolescent pregnancy. However, differences in race, income, or geographic areas among mental disorder types may exist, which can affect the estimation of adolescent pregnancy. Future study that provide a basis for adjusting for differences in the age-specific adolescent pregnancy rates across groups defined by sex, race, ethnicity, geography, and other sociodemographic categories are warranted.”

Results:

1. Major: Terminology about the study population can be strengthened. Use of the term, ‘girls’ vs ‘females’ vs ‘adolescents’ vs ‘women’ makes it hard to follow. Please pick one and use it consistently throughout. ‘Adolescence’ may be the most appropriate and gender-neutral descriptor. Alternatively, you can make a note in the introduction describing that you will use the terms interchangeably.

Thank you for your suggestion. We have carefully revised the manuscript to use adolescent consistently as the term to describe the study population. 

2. Minor: Please define a tic disorder?

Below are the ICD-9-CM codes that this study used to define a tic disorder. 

307.20: Tic disorder, unspecified

307.21: Transient tic disorder

307.22: Chronic motor or vocal tic disorder

The above information is available in eTable 2. Based on CCS classification, all of the three ICD-9-codes are classified as Disorders Usually Diagnosed In Infancy, Childhood, or Adolescence.

3. Minor: Please include confidence intervals when presenting your data.

We included 95% confidence intervals throughout the manuscript whenever we saw fit. 

4. Minor: Which exposure group does the 7.4 per 1000 and 6.5 per 1,000 finding pertain to? (Page 5)

The answers are as follow: Developmental Disorders (7.4 events/ per 1,000 person-years, 95% CI: 6.1 – 8.7) and Disorders Usually Diagnosed in Infancy, Childhood, or Adolescence (DICA), such as autism spectrum disorders or tic disorders (6.5 events/ per 1,000person-years, 95% CI: 5.4 – 7.6).

5. Major: Please save all language that is geared towards interpreting the results for the discussion section (ex: ‘Consistent with previous studies’, ‘as expected’).

We removed ‘Consistent with previous studies’ and ‘as expected’ from the manuscript. 

6. Major: Comorbidities are presented in the analyses and therefore should be discussed in the methods.

We have added the following paragraph in the methods section to explain how we addressed the concurrent mental disorders in the statistical analyses. (Lines 150 – 158)

“Finally, multivariable logistic regressions were used to identify the relationships between the types of mental disorders and the incidence of adolescent pregnancy stratified by age. In the logistic regression model, a binary coding approach was used to create separate dummy variables for each mental disorder category. This was used to assess the individual effects of each disorder while accounting for potential overlapping effects. However, it is important to note that the model did not explicitly consider the interactions or combined effects between multiple concurrent mental disorders. Instead, the focus was solely on examining the main effects of mental disorder variables on pregnancy events.”

7. Minor: A hypothesis was not presented originally in the introduction and therefore ‘as expected’ is not appropriate in this section nor throughout the manuscript.

We removed the phrase ‘as expected’ from the manuscript.

8. Major: The organization of study findings can be revised dramatically to increase readability. Please present prevalence’s and then measures of effect. As written, the outcome is unclear.

We have re-organized the manuscript to present the pregnancy incidence findings, followed by the measure of effects as the signals of association between mental disorder type and adolescent pregnancy. 

9. Minor: Physician-diagnosed mental disorders is used for the first time here. Either make consistent throughout or remove from this section.

We noticed that we used mental disorders, diagnosed mental disorders, and physician-diagnosed mental disorders interchangeably throughout the manuscript, which may lead to confusion. As the reviewer suggested, we have revised the manuscript and use “diagnosed mental disorder” consistently throughout the text.

10. Table 2: The exposure group of ‘adolescents with at least one mental disorder diagnosis’ is clear. This is the language that should be used throughout.

We agree that “adolescents with at least one mental disorder diagnosis” is the operational definition used in our study to define adolescents with a pre-existing mental disorder diagnosis. Thus, we revised the text and used the operational definition as we saw fit. 

11. Table 2: Please round to 2 decimal places.

We modified table 2 as suggested. 

Discussion:

1. Major: It is unclear what the intent of paragraph 2 is, and I am not certain that the implications described here are justified based on the findings.

We have removed the following paragraph from the manuscript. 

“Pregnancy can result in both live births and non-live births (e.g., abortions). The trends in the adolescent pregnancy rate between 1973 and 2017 are available and were reported by the Guttmacher Institute. The decline in the adolescent pregnancy rate over the past two and a half decades was reflected by declines in both birth and abortion rates; however, we noted that the pregnancy rate did not change significantly among the age group under 15 years. Together with our study findings, additional efforts to decrease pregnancy rates among sub-populations in adolescents who may be at a higher risk for unplanned pregnancy, individuals who are considered too young to have children and/or who have mental disorders are required.”

2. Minor: Some of the articles chosen to compare study findings may need revision. For this manuscript, I do not think treatment seeking women with opioid use disorder is a good group to use to present results in light of other work.

We removed the citation (the study on women with OUD seeking treatment). 

3. Major: This section should be revised to avoid stigmatizing language and to more aligned with harm reduction approaches. Intervention approaches should also be mindful of this unique age group. Particularly on page 3 use of the term ‘healthy peers’ is very jarring and should not be used at all.

We revised the sentence as below. (Lines 284 – 286)

“Further evaluation of the pregnancy intentions of adolescents with ID will be particularly meaningful given the known vulnerability of children with ID in becoming victims of sexual abuse.”

4. Minor: What does DICA stand for?

DICA stands for “Disorders usually diagnosed in Infancy, Childhood, or Adolescence (DICA)” and was first introduced in the Results section.

5. Major: Again, I think before the discussion can be framed around unplanned pregnancy the introduction needs to be clear about this as the goal of this investigator. Also, while pregnancy intent is not something that could possibly be obtained from the data source, it may be helpful to create a directed acyclic graph to make sense of some of these relationships. If anything, it may be that unplanned pregnancy is a mediator.

We have edited the introduction to inform readers that adolescent pregnancy can be intended and unintended. With supporting information on the potential effect of mental disorders on pregnancy intention and unintended pregnancy, we tried to frame the manuscript around unintended pregnancy. As suggested by the reviewer, we have included a directed acyclic graph in the discussion to illustrate the relationship between mental disorder, pregnancy intention and adolescent pregnancy. 

6. Minor: STI’s do not contribute to unplanned pregnancy (page 4).

The following sentence has been removed from the manuscript to address one of your previous comments. “For example, mental disorders have been associated with the increased likelihood of being sexually active at a younger age, higher rates of sexually transmitted diseases, and becoming victims of sexual abuse during childhood, indicating that adolescents with mental disorders are at a higher risk of unplanned pregnancy. (27, 32, 33)” 

7. Minor: Please be consistent with what is the outcome vs outcome type throughout.

We chose to use outcome type and made the appropriate changes as in the manuscript. 

Reviewer #2: Little is known about the association between mental health and adolescent pregnancy. In this manuscript, the authors report on the findings of a retrospective case-cohort study of females at age 13-19. This study, therefore, makes a valuable contribution to the literature. Below I give some suggestions to improve the paper.

• It is unclear how the retrospective cohort of females was established (first sentence under methods). Perhaps it would be better to describe the source of data, and then follow this up with the description of the retrospective cohort. In describing the cohort, the text in the parentheses (i.e., eligibility period…..) in the first sentence can be deleted.

Thank you for your suggestion. We have made changes to first describe the data source and then describe the retrospective cohort. (Lines 75 – 85)

“Data source 

This study used healthcare claims data obtained from the Merative™ MarketScan Commercial Claims Research Database. This database provides details on reimbursed health services, including medical encounters and drugs dispensed in outpatient pharmacies for patients in approximately 150 employer-sponsored insurance plans. Because the database includes claims from many private insurers and has very wide geographic coverage, it has been frequently used in the analysis of healthcare utilization as a data source representing privately insured individuals. (12-14) A retrospective cohort of females aged 13 - 19 years was established between 2006 and 2014. The inclusion criteria for the cohort was as follows: continuous enrollment for at least three years between 2005 and 2015, allowing for eligibility periods such as 2005-2007, 2006-2008 and 2013-2015.”

• It is unclear if the database includes information on sociodemographic characteristics of privately insured people (e.g., race, income, etc.). Including these in the model would be useful in assessing whether these characteristics are also associated with the outcomes. These data would also be useful in understanding how the study sample may differ from the wider population. This could also be highlighted as a limitation if the data are not available.

We acknowledge the limitation that we only included “age” in our analysis as opposed to other sociodemographic information available in the data source. We added the following paragraph to address the limitation in the discussion section. (Lines 305 – 311)

“Fourth, we included age as the only covariate in the multivariable logistic regression after adjusting for the relationship between the type of mental disorder and adolescent pregnancy. However, differences in race, income, or geographic areas among mental disorder types may exist, which can affect the estimation of adolescent pregnancy. Future study that provide a basis for adjusting for differences in the age-specific adolescent pregnancy rates across groups defined by sex, race, ethnicity, geography, and other sociodemographic categories are warranted.”

• It would have been useful to show whether the likelihood of pregnancy was different for those with comorbid mental health disorders compared to those with only one mental disorder diagnosis. This could have implications for the targeting of prevention interventions.

We agree that 1) mental disorders often co-occur, and 2) the likelihood of pregnancy could be different for those with comorbid mental disorders compared to those with only one mental disorder. In our logistic regression model, we did not specifically account for patients with multiple concurrent mental disorders. Rather, we treated each mental disorder as a separate independent variable. Therefore, the interpretation of the model for a patient with multiple concurrent mental disorders would involve examining the individual effects of each disorder independently, without explicitly considering potential interactions or combined effects. To clarify, we have added the following paragraph. (Lines 152 – 158)

“In the logistic regression model, a binary coding approach was used to create separate dummy variables for each mental disorder category. This was used to assess the individual effects of each disorder while accounting for potential overlapping effects. However, it is important to note that the model did not explicitly consider the interactions or combined effects between multiple concurrent mental disorders. Instead, the focus was solely on examining the main effects of the mental disorder variables on pregnancy events.”

Minor edits

• The second paragraph in the discussion does not seem to add much and can be deleted.

The second paragraph in the discussion section was deleted.

• The manuscript needs a careful read through for grammar and spelling errors. Below I highlight just a few from the first two paragraphs

- Para 1, First sentence: Mental health disorders in “adolescence”

- Para 1, Second sentence: “The 2020 national prevalence…. in high schools in the United States was 42%”

- Para 1, Third sentence: “hopelessness has increased dramatically”

- Para 2, First sentence: “adolescent mental”

- Para 2, Second sentence: “An existing mental health problem/disorder”

Thank you so much for identifying the grammar and spelling errors listed above. We have carefully proofread the manuscript to correct not only the errors listed above, but also additional errors throughout the revised manuscript.

---

## [Decision Letter · Decision Letter 1]

19 Sep 2023

PONE-D-23-05808R1Comparing pregnancy and pregnancy outcome rates between adolescents with and without pre-existing mental disordersPLOS ONE

Dear Dr. Jeon,

Thank you for submitting your manuscript to October 9. After careful consideration, we feel that it has merit but does not fully meet PLOS ONE’s publication criteria as it currently stands. Therefore, we invite you to submit a revised version of the manuscript that addresses the points raised during the review process.

We look forward to receiving your revised manuscript.

Kind regards,

Giuseppe Marano

Academic Editor

PLOS ONE

Journal Requirements:

Reviewers' comments:

Reviewer's Responses to Questions

**Comments to the Author**

1. If the authors have adequately addressed your comments raised in a previous round of review and you feel that this manuscript is now acceptable for publication, you may indicate that here to bypass the “Comments to the Author” section, enter your conflict of interest statement in the “Confidential to Editor” section, and submit your "Accept" recommendation.

Reviewer #1: All comments have been addressed

Reviewer #2: (No Response)

2. Is the manuscript technically sound, and do the data support the conclusions?

Reviewer #1: Yes

Reviewer #2: Yes

3. Has the statistical analysis been performed appropriately and rigorously? 

Reviewer #1: Yes

Reviewer #2: I Don't Know

4. Have the authors made all data underlying the findings in their manuscript fully available?

Reviewer #1: Yes

Reviewer #2: Yes

5. Is the manuscript presented in an intelligible fashion and written in standard English?

Reviewer #1: Yes

Reviewer #2: Yes

6. Review Comments to the Author

Reviewer #1: Thank you for being responsive to our review of your manuscript. This version has been carefully edited and therefore strongly enhanced both in terms of readability and in scientific reproducibility.

Reviewer #2: The authors have done a great job responding to earlier comments.

In response to earlier comments, the authors included a paragraph in the introduction (line 57 - 66) that I feel is unnecessary given that the data used do not provide details on whether pregnancies were intended or unintended. The authors already clearly articulate the lack of data on pregnancy as a limitation. What they have written in the discussion is adequate.

There are some minor grammatical errors that I have highlighted in the attached document in tracked changes

7. PLOS authors have the option to publish the peer review history of their article (what does this mean?). If published, this will include your full peer review and any attached files.

Reviewer #1: **Yes: **Brian W. Jack

Reviewer #2: No

---

## [Author Response · Author response to Decision Letter 1]

26 Sep 2023

We reviewed the reference list to ensure it is complete and correct. Also, there were no references that have been retracted. 

References #11, #14, #20, #21, and #32 were re-formatted to meet the submission guideline. (See below). As for the reference #21, we have updated the URL as the address has been changed since we checked it last time. 

11. Sekharan VS, Kim TH, Oulman E, Tamim H. Prevalence and characteristics of intended adolescent pregnancy: an analysis of the Canadian maternity experiences survey. Reprod Health. 2015;12:101.

14.Merative™ MarketScan Research Databases. [Cited 2023 Sep 26]. Available from: https://www.merative.com/real-world-evidence. 

20.Files H. Clinical Classifications Software (CCS) for ICD-9-CM: Agency for healthcare research and quality. [Cited 2023 Sep 26]. Available from: https://www.hcup-us.ahrq.gov/toolssoftware/ccs/ccs.jsp.

21.Centers for Disease Control and Prevention. Annual Estimates of the resident population by single year of age and sex for the United States. U.S. Department of Health & Human Services. [Cited 2023 Sep 26]. Available from: https://catalog.data.gov/dataset/us-census-annual-estimates-of-the-resident-population-for-selected-age-groups-by-sex-for-t. 

32.Kost K M-ZI, Arpaia A. Pregnancies, births and abortions among adolescents and young women in the United States: Guttmacher Institute; 2013. [Cited 2023 Sep 26]. Available from: https://www.guttmacher.org/sites/default/files/report_pdf/us-adolescent-pregnancy-trends-2013.pdf.

---

## [Decision Letter · Decision Letter 2]

18 Oct 2023

PONE-D-23-05808R2Comparing pregnancy and pregnancy outcome rates between adolescents with and without pre-existing mental disordersPLOS ONE

Dear Dr. Jeon,

Thank you for submitting your manuscript to PLOS ONE. After careful consideration, we feel that it has merit but does not fully meet PLOS ONE’s publication criteria as it currently stands. Therefore, we invite you to submit a revised version of the manuscript that addresses the points raised during the review process.

We look forward to receiving your revised manuscript.

Kind regards,

Giuseppe Marano

Academic Editor

PLOS ONE

Journal Requirements:

Reviewers' comments:

Reviewer's Responses to Questions

**Comments to the Author**

1. If the authors have adequately addressed your comments raised in a previous round of review and you feel that this manuscript is now acceptable for publication, you may indicate that here to bypass the “Comments to the Author” section, enter your conflict of interest statement in the “Confidential to Editor” section, and submit your "Accept" recommendation.

Reviewer #1: (No Response)

Reviewer #2: (No Response)

2. Is the manuscript technically sound, and do the data support the conclusions?

Reviewer #1: Yes

Reviewer #2: Yes

3. Has the statistical analysis been performed appropriately and rigorously? 

Reviewer #1: Yes

Reviewer #2: I Don't Know

4. Have the authors made all data underlying the findings in their manuscript fully available?

Reviewer #1: Yes

Reviewer #2: Yes

5. Is the manuscript presented in an intelligible fashion and written in standard English?

Reviewer #1: Yes

Reviewer #2: Yes

6. Review Comments to the Author

Reviewer #1: The authors have responded to suggestions

Reviewer #2: The comment in the introduction is still unanswered.

In the introduction, line 44: confirm if the inserted word, "have", is the right one and delete the question mark

7. PLOS authors have the option to publish the peer review history of their article (what does this mean?). If published, this will include your full peer review and any attached files.

Reviewer #1: No

Reviewer #2: No

---

## [Author Response · Author response to Decision Letter 2]

3 Dec 2023

Reviewer #2 Comment: In response to earlier comments, the authors included a paragraph in the introduction (line 57 - 66) that I feel is unnecessary given that the data used do not provide details on whether pregnancies were intended or unintended. The authors already clearly articulate the lack of data on pregnancy as a limitation. What they have written in the discussion is adequate.

The lines 57-66 are now removed and along with the removal, the ref 10 is relocated to ref 30 and ref 11 is deleted. In addition, we updated the reference numbering as follow;

- References 12-31 are renumbered to ref 10-29 

- References 32&33 are now ref 31&32

---

## [Decision Letter · Decision Letter 3]

14 Dec 2023

Comparing pregnancy and pregnancy outcome rates between adolescents with and without pre-existing mental disorders

PONE-D-23-05808R3

Dear Dr. Jeon,

We’re pleased to inform you that your manuscript has been judged scientifically suitable for publication and will be formally accepted for publication once it meets all outstanding technical requirements.

Kind regards,

Giuseppe Marano

Academic Editor

PLOS ONE

Additional Editor Comments (optional):

Reviewers' comments:

Reviewer's Responses to Questions

**Comments to the Author**

1. If the authors have adequately addressed your comments raised in a previous round of review and you feel that this manuscript is now acceptable for publication, you may indicate that here to bypass the “Comments to the Author” section, enter your conflict of interest statement in the “Confidential to Editor” section, and submit your "Accept" recommendation.

Reviewer #1: All comments have been addressed

Reviewer #2: All comments have been addressed

2. Is the manuscript technically sound, and do the data support the conclusions?

Reviewer #1: Yes

Reviewer #2: Yes

3. Has the statistical analysis been performed appropriately and rigorously? 

Reviewer #1: Yes

Reviewer #2: I Don't Know

4. Have the authors made all data underlying the findings in their manuscript fully available?

Reviewer #1: Yes

Reviewer #2: Yes

5. Is the manuscript presented in an intelligible fashion and written in standard English?

Reviewer #1: Yes

Reviewer #2: Yes

6. Review Comments to the Author

Reviewer #1: Thank you for your responsiveness to reviewer comments and for this important contribution to the field.

Reviewer #2: (No Response)

7. PLOS authors have the option to publish the peer review history of their article (what does this mean?). If published, this will include your full peer review and any attached files.

Reviewer #1: No

Reviewer #2: No

---

## [Editor Report · Acceptance letter]

4 Mar 2024

PONE-D-23-05808R3 

PLOS ONE

Dear Dr. Jeon, 

I'm pleased to inform you that your manuscript has been deemed suitable for publication in PLOS ONE. Congratulations! Your manuscript is now being handed over to our production team.

Kind regards, 

on behalf of

Dr. Giuseppe Marano 

Academic Editor

PLOS ONE